# Preventing Sepsis in Preterm Infants with Bovine Lactoferrin: A Randomized Trial Exploring Immune and Antioxidant Effects

**DOI:** 10.3390/nu17193154

**Published:** 2025-10-03

**Authors:** Virginia Plaza-Astasio, Belén Pastor-Villaescusa, Mª Cruz Rico-Prados, María Dolores Mesa-García, María José Párraga-Quiles, María Dolores Ruiz-González, Pilar Jaraba-Caballero, Inés Tofé-Valera, María José de la Torre-Aguilar, María Dolores Ordóñez-Díaz

**Affiliations:** 1University of Córdoba, Metabolism and Investigation Unit, Maimonides Institute of Biomedicine Research of Córdoba (IMIBIC), Reina Sofia University Hospital, 14004 Córdoba, Spain; h42plasv@uco.es (V.P.-A.); belen.pastor@imibic.org (B.P.-V.); mdordonezdiaz@gmail.com (M.D.O.-D.); 2Neonatology Unit, Reina Sofia University Hospital, 14004 Córdoba, Spain; mjose.parraga.sspa@juntadeandalucia.es (M.J.P.-Q.); mariolaruiz11@gmail.com (M.D.R.-G.); pjaraba@gmail.com (P.J.-C.); itofevalera3@gmail.com (I.T.-V.); 3Primary Care Interventions to Prevent Maternal and Child Chronic Diseases of Perinatal and Developmental Origin (RICORS), RD21/0012/0008, Instituto de Salud Carlos III, 28029 Madrid, Spain; mdcrico@ugr.es (M.C.R.-P.); mdmesa@ugr.es (M.D.M.-G.); 4Network in Maternal, Neonatal, Child and Developmental Health Research (RICORS-SAMID, RD24/0013/0007) Instituto de Salud Carlos III, 28029 Madrid, Spain; 5Department of Biochemistry and Molecular Biology II, School of Pharmacy, Institute of Nutrition and Food Technology “José Mataix”, Biomedical Research Center, University of Granada, 18016 Granada, Spain; 6Instituto de Investigación Biosanitaria de Granada (ibs.GRANADA), 18016 Granada, Spain; 7Endocrinology and Child Nutrition Unit, Maimonides Institute of Biomedicine Research of Córdoba (IMIBIC), Reina Sofia University Hospital, University of Córdoba, 14004 Córdoba, Spain

**Keywords:** infant, premature, sepsis, interleukins, antioxidants, hemoglobin, iron

## Abstract

**Background/Objectives**: Late-onset neonatal sepsis (LOS) remains a leading cause of morbidity and mortality in very low birth weight (VLBW) infants (<1500 g and/or gestational age <32 weeks), with limited preventive strategies. We evaluated whether early enteral bovine lactoferrin (bLf), given its antimicrobial, immunomodulatory, and antioxidant properties, reduces LOS and improves immunologic, antioxidant, and hematologic markers in these infants. **Methods**: In this randomized, double-blind, placebo-controlled trial, 103 VLBW infants received bLf (150 mg/kg/day; *n* = 50) or the placebo (*n* = 53) within 72 h of birth for four weeks or until discharge. Outcomes included culture-confirmed LOS, mortality, and major morbidities. Risk ratios (RRs) were calculated, adjusting for gestational age, human milk intake, and ventilatory support when ≥25 events occurred. Pre/post changes in cytokines, total antioxidant capacity (TAC), and hemoglobin (Hb) were analyzed for interaction effects (time x intervention). **Results**: bLf reduced LOS (adjusted RR 0.54; 95% CI 0.31–0.93; *p* = 0.028), without differences in other morbidities or mortality. bLf preserved MCP-1 levels, declining in the placebo group (interaction *p* = 0.022). Among LOS infants receiving bLf, IL-6 remained stable and MCP-1 increased, while both declined in other groups (interaction *p* = 0.007 for IL-6; *p* = 0.052 for MCP-1). Although TAC showed a non-significant interaction, the placebo group declined (*p* = 0.002), while bLf remained stable (*p* = 0.400) in the post hoc analysis. In non-transfused infants, bLf increased Hb by 0.9 g/dL vs. controls (*p* = 0.028). **Conclusions**: Early bLf supplementation safely reduces LOS in VLBW infants and may support immunologic, antioxidant, and hematologic stability.

## 1. Introduction

Late-onset neonatal sepsis (LOS) is a significant cause of morbidity and mortality in preterm infants, particularly among those with very low birth weight (VLBW, ≤1500 g) [1,2]. The incidence of LOS in VLBW infants ranges from 18% to 26%, rising to nearly 39% in those with extremely low birth weight (ELBW, ≤1000 g) [3]. Beyond its impact on mortality, LOS is associated with complications such as necrotizing enterocolitis (NEC), bronchopulmonary dysplasia (BPD), and retinopathy of prematurity (ROP). These complications contributed to prolonged hospital stays, increased healthcare costs, and long-term adverse effects on neurocognitive development and quality of life among survivors [4]. Collectively, these observations underscore the critical need for effective preventive strategies to prevent LOS. In this context, lactoferrin (Lf), a glycoprotein present in human milk, has garnered interest due to its antimicrobial, immunomodulatory, antioxidant, anti-inflammatory, bifidogenic, and neuroprotective properties [5]. In VLBW infants, who often require several days to achieve full enteral feeding, the naturally available concentration of Lf may be insufficient, suggesting that early enteral supplementation could reduce the risk of LOS.

Both human and bovine lactoferrin (hLf/bLf) inhibit bacteria via iron sequestration and direct membrane/lipopolysaccharide (LPS) interaction and share the same N-terminal LPS-binding region. Upon pepsin digestion, N-terminal fragments, such as lactoferricin, are released. In vitro, bovine-derived peptides exhibit minimum inhibitory concentrations four times lower than their human analogue, with activity being highest in the apo form and modulated by divalent cations, although effects are strain- and context-dependent [6,7]. Beyond its similarity to hLf, bLf is manufactured on an industrial scale under Current Good Manufacturing Practice (cGMP), and its use in foods is supported by regulatory assessments from the U.S. Food and Drug Administration (FDA; GRAS ‘no-questions’) and the European Food Safety Authority (EFSA; novel-food opinions). Moreover, bLf is more readily available commercially and less costly than recombinant hLf, making it the primary option for investigation and nutritional supplementation, especially in preterm infants [8].

The first clinical trial assessing bLf supplementation in VLBW infants, conducted by Manzoni et al. in 2009 [9], showed a marked decline in the incidence of LOS among infants receiving bLf, either alone or combined with *Lactobacillus rhamnosus* (LGG). This reduction was especially notable in VLBW infants and cases of LOS caused by Gram-negative bacteria and *Candida* spp. In addition, the incidence of NEC at stage ≥ 2 (in the bLf and LGG group), severe ROP, and sepsis-attributable mortality also declined. Since then, numerous comprehensive studies evaluating the potential protective effect of bLf have been published [10,11,12,13,14,15,16,17,18,19]. In a 2020 systematic review, Pammi and Gautham [20] concluded that bLf, with or without LGG, may reduce the incidence of LOS and the duration of hospital stay in preterm infants, without evidence of significant adverse effects. However, due to heterogeneous study designs and methodological limitations, the quality of the evidence was rated as low. This highlights the need for well-designed, high-quality trials to draw robust conclusions, define the optimal dosage and duration of bLf supplementation, and assess long-term effects on neurocognitive outcomes.

Beyond its role in sepsis prevention, bLf may enhance hemoglobin levels by promoting iron homeostasis and enhancing absorption, offering potential clinical benefits in the management of anemia of prematurity [21]. Notably, a study by Abd Elrahman et al. [22] demonstrated that preterm infants receiving bLf at a dose of 100 mg/day from the first week of life exhibited significantly higher hemoglobin levels compared to those who did not receive supplementation. Furthermore, although clinical evidence remains limited and inconclusive, promising findings suggest that bLf may exert additional benefits through its immunomodulatory, anti-inflammatory, antioxidant, neuroprotective, and bifidogenic properties [23,24]. In this context, Akin et al. [19] demonstrated that bLf supplementation not only reduced LOS and NEC but also led to a 43% increase in regulatory T cell (FOXP3-CD4+CD25) expression. The authors suggested that this immunologic shift may help lower the risk of infections and intestinal inflammation. However, the small sample size and the lack of direct quantification of intestinal cellular responses limited the strength of these conclusions. Regarding its antioxidant potential, in vivo studies have shown that Lf supplementation decreased lipid peroxidation and reactive oxygen species production in the presence of iron. Nonetheless, the clinical relevance of these findings remains unclear [25,26].

Given this context, the present study aimed to evaluate the efficacy of enteral bLf supplementation in reducing overall LOS, mortality, sepsis-related mortality, and major neonatal complications (such as NEC, ROP, BPD, hemodynamically significant patent ductus arteriosus (hsPDA), and brain injury identified at discharge) in VLBW and/or ≤32-week-old GA preterm infants. The study also aimed to explore the underlying mechanisms of bLf’s potential benefits by investigating immunological and antioxidant markers. Furthermore, we also assessed whether the bLf administration could offer further clinical advantages, such as improving hemoglobin levels.

## 2. Materials and Methods

### 2.1. Study Design

This randomized, double-blind, parallel, placebo-controlled clinical trial was conducted in the tertiary Neonatal Intensive Care Unit (NICU) at Reina Sofía University Hospital (HURS) in Córdoba (Spain), in collaboration with the Maimonides Biomedical Research Institute of Córdoba and the Institute of Nutrition and Food Technology of Granada (INYTA, Granada, Spain).

This study adhered to the Declaration of Helsinki and received approval from the Institutional Hospital Ethical Committee (ethics approval number: 3335; approval date: 28 November 2016). Parents provided written informed consent prior to enrolment. Data were recorded in an anonymized database in compliance with Spain’s Organic Law 15/1999 of December 13 on the Protection of Personal Data (updated by Organic Law 3/2018 of December 5 on Personal Data Protection and Guarantee of Digital Rights). The trial was prospectively registered on ClinicalTrials.gov (NCT03472170). The Consolidated Standards of Reporting Trials statement (CONSORT) was followed in the reporting of the study design and results, abstract, and flow diagram (Appendix A).

### 2.2. Study Participants

Preterm infants were recruited from November 2017 to April 2020. The inclusion criteria were: VLBW (≤1500 g) and/or a GA ≤ 32 weeks, and age < 72 h at randomization. Exclusion criteria included early-onset sepsis, congenital gastrointestinal anomalies, chromosomal disorders, congenital anomalies or genetic conditions with no survival expectancy, and severe perinatal hypoxia.

### 2.3. Randomization and Bias Control

Eligible infants were randomly allocated in a 1:1 ratio to either the intervention group (bLf) or the control group (placebo) using a randomization list generated by the statistical program Epidat (Version 4.1, October 2014, Xunta de Galicia, Santiago de Compostela, Spain). The trial employed a double-blind design, ensuring that neither the infant’s caregivers nor the researchers knew the composition of the administered product. To maintain blinding, the hospital pharmacy was responsible for preparing the placebo, ensuring that it had identical packaging and organoleptic characteristics to the active product.

An intention-to-treat (ITT) analysis was conducted to minimize potential biases resulting from participant withdrawal after randomization. Additional measures to control bias include standardizing clinical criteria for newborn inclusion and ensuring a rigorous and homogeneous selection process. Furthermore, the study conditions were maintained consistently throughout the trial to avoid any variations that could compromise the validity of the data collected.

### 2.4. Intervention

The bLf-based nutritional supplement was a product commercialized (Dicofarm^®^, Rome, Italy) in compliance with European Union regulations. It was approved by the European Food Safety Authority (EFSA) in 2012 and by the U.S. Food and Drug Administration (FDA) in 2013. The hospital pharmacy department prepared both the bLf supplement and the placebo, adhering to an established dosage of 150 mg/kg/day (up to a maximum of 300 mg/day). The placebo was formulated to match the bLf supplement in appearance and taste. Both preparations were administered enterally in liquid form, either orally or via a nasogastric tube, using the smallest volume possible. Supplementation began within the first 72 h of life, regardless of the volume of enteral feeds achieved, and was also given during periods of nil per os (NPO). It was continued once daily for up to four weeks or until hospital discharge to home, whichever occurred first. Daily administration was recorded by the nurse, documenting any associated incident. Our NICU followed a standardized feeding protocol based on the current recommendations of the European Society for Paediatric Gastroenterology, Hepatology and Nutrition and the Spanish Society for Neonatology on nutrient intakes and nutritional management for preterm infants. The nutritional protocol included: (1) Administration of standardized parenteral nutrition with all macronutrients within the first hours of life; (2) Early trophic enteral feedings; (3) Gradual advancement of enteral volumes; (4) Preference for mother’s own milk; donor milk was not available during the study period, and formula was used when necessary; (5) Fortification of breastfeeding starting at an approximate volume of 80 mL/kg/d; (6) Discontinuation of parenteral nutrition when enteral nutrition reached a volume of 100–120 mL/kg/d for two days. Probiotics were not administered.

### 2.5. General Characteristics and Clinical History

Relevant maternal and neonatal data were prospectively collected from medical records. Maternal variables included age (years), presence of preeclampsia, diabetes, and chorioamnionitis, administration of a complete course of antenatal corticosteroids, antibiotic use, and vertical sepsis risk factors defined by the Disease Control and Prevention (CDC) perinatal guidelines (prolonged rupture of membranes > 18 h, intrapartum fever ≥ 38 °C, or evidence of β-hemolytic Streptococcus, either vaginal colonization or a positive urine culture) [27].

Neonatal data included GA (weeks), sex, birth weight (g; z-scores), birth length (cm; z-scores), and head circumference (cm; z-scores). Documented interventions comprised surfactant administration, need for endotracheal intubation, and duration (≥48 h according to the CDC and National Nosocomial Infections Surveillance (NNIS) criteria, reflecting the typical onset of ventilator-associated pneumonia [28] and significant endotracheal tube colonization [29,30], total duration of respiratory support (days), receipt of blood product transfusions, with the binary variable red blood cell transfusion (RBCT) status, and use of postnatal antibiotics or proton pump inhibitors (PPIs).

Nutritional variables included the duration of parenteral nutrition administration, the time to first enteral feed, and the time to achieve full enteral feeding. The proportion of infants receiving full enteral feeding with their own mother’s milk or a combination of breast milk and formula was also recorded. Discharge data comprised the total hospital stay (in days) and anthropometric measurements (weight, length, and head circumference), each expressed as both absolute values and z-scores.

### 2.6. Outcome Measures

#### 2.6.1. LOS

According to the NeoKiss Surveillance Manual (Version 2, 2015) [31], LOS incidence was defined as the first laboratory-confirmed sepsis episode occurring after 72 h of life.

#### 2.6.2. Neonatal Complications, Overall Mortality, and Mortality Attributable to LOS

The diagnostic criteria for each condition are detailed in Table 1 (please note: the following references are cited in Table 1: [32,33,34,35]).

#### 2.6.3. Biochemical, Inflammation, and Antioxidant Markers

Samples were collected at two time points. The first sample was obtained prior to administering either bLf or placebo, and the post-intervention sample was drawn within 48 h of concluding the assigned supplementation, irrespective of whether the entire regimen was completed. At each point, 3 mL of venous blood was drawn, 1.5 mL into an EDTA tube (for complete blood count and plasma) and 1.5 mL into a gel barrier tube (for serum). Samples were centrifuged at 3000–3500 rpm (4 °C, 10 min) to separate plasma and serum. Erythrocytes were washed 2–3 times with an isotonic saline solution to remove any residual plasma and then aliquoted. Plasma, serum, and erythrocyte aliquots were stored at –80 °C until analysis.

Standard biochemical and hematological parameters were measured in the hospital laboratory using standardized colorimetric and enzymatic methods, with strict quality control protocols.

Inflammatory markers relevant to neonatal immune response, such as interleukins (IL-1β, IL-6, IL-8, IL-10), tumor necrosis factor-α (TNF-α), interferon-γ (IFN-γ), and monocyte chemoattractant protein-1 (MCP-1), were quantified in plasma using the Luminex xMAP technology (HADK2MAG-61K kit, Millipore Corporation, Darmstadt, Germany), and finally expressed as pg/mL.

Antioxidant defense was assessed both enzymatically (in erythrocytes) and non-enzymatically (in serum). Due to an incident during blood collection, erythrocyte samples for antioxidant enzyme analyses were available for only 46 newborns. Hemoglobin concentration in the erythrocyte lysates was determined using the DetectX Hemoglobin Detection Kit (Arbor Assays, Ann Arbor, MI, USA), which converts hemoglobin into cyanmethemoglobin. Enzymatic activity for catalase (DetectX Catalase Colorimetric Activity Kit, London, UK), glutathione peroxidase (EnzyChrom™ Glutathione Peroxidase Assay Kit, San Jose, CA, USA), glutathione reductase (EnzyChrom™ Glutathione Reductase Kit), and superoxide dismutase (EnzyChrom™ Superoxide Dismutase Assay Kit) was measured spectrophotometrically and expressed as specific activity units per gram of hemoglobin (U/g Hb). For total antioxidant capacity (TAC) in serum, samples were analyzed with the Cayman Chemical Antioxidant Assay Kit, Ann Arbor, MI, USA, which measures the capacity to inhibit ABTS oxidation in the presence of metmyoglobin. Trolox was used as the reference standard, and TAC values were expressed as millimolar Trolox equivalents (mmol Trolox/L).

### 2.7. Sample Size

Sample size was calculated a priori using GRANMO software, version 7.12, (Institut Municipal d’Investigació Mèdica, Barcelona, Spain). Surveillance data from our unit over the two years preceding study initiation (SEN 1500 registry) indicated a 40% incidence of LOS. Drawing on previous RCTs of bLf, which reported relative LOS reductions of 45–66% [7,12], we powered the study to detect a 60% decrease (from 40% to 16%). With a two-tailed α = 0.05, 80% power, and allowance for 5% attrition, 66 infants per arm were required (total N = 132). However, COVID-19 caused unavoidable recruitment interruptions and changes in care pathways, limiting enrollment to 103 preterm infants (50 in the bLf arm and 53 in the placebo arm). A post hoc power analysis indicated approximately 70% power to detect the estimated effect size.

### 2.8. Statistical Analysis

For quantitative variables, data were reported as mean and standard deviation (SD) or median and interquartile range (IQR), depending on the data distribution, which was tested for normality using Q–Q plots and histograms. Qualitative variables were described in terms of counts and percentages (*n*, %).

Comparisons of quantitative data between groups were performed using Student’s *t*-tests for normal distributions or Mann–Whitney U tests for non-normal distributions. Categorical variables were analyzed using Chi-square (χ^2^) tests, Fisher’s exact test (when one or more expected cell counts were <5) or Fisher–Freeman–Halton (2 × 3).

The disease incidence among participants in each group was initially expressed as crude risk ratio (RR, 95% confidence interval (CI)), calculated from 2 × 2 contingency tables. For variables with ≥25 total events [36] (LOS and BPD), adjusted RR (aRR, along with 95% CI, and number needed to treat (NNT) when appropriate) was estimated using modified Poisson regression with a log link and robust (sandwich) variance estimation [37]. Models were estimated to include treatment group (bLf vs. placebo), GA, and feeding type during the trial (mixed feeding vs. formula only). Additionally, the LOS model included intubation ≥ 48 h, while the BPD model included total days of invasive respiratory support (excluding those associated with surgical or other non-respiratory causes) until BPD diagnosis criteria were met. Infant sex was not included, as it was neither significant nor confounding and led to convergence issues.

A mixed factorial analysis of variance (ANOVA) was performed to evaluate changes in inflammatory and antioxidant markers over time (pre- vs. post-intervention) and between intervention arms (bLf vs. placebo). Missing values due to undetectable concentrations were imputed using half the minimum positive value recorded for each variable. Data were then log-transformed (log10) to meet the assumptions of normality, homogeneity of variances, and homogeneity of covariances. Pairwise comparisons were adjusted using the Bonferroni correction. When baseline values significantly differed between intervention groups, analysis of covariance (ANCOVA) was applied. For clarity and ease of interpretation, results are presented using untransformed values expressed as medians and IQR. Additional analyses incorporated LOS as a between-subjects factor to further explore the data, assessing its interaction with intervention and time. The same statistical procedures and adjustments were applied in these models.

A mixed factorial ANOVA was also conducted to assess hemoglobin levels, with time (pre-intervention and post-intervention) as a within-subject factor and intervention group (bLf vs. placebo) and transfusion status (yes vs. no) as between-subject factors. The same statistical checks described above were applied.

*p* values < 0.05 were considered statistically significant. Data analyses were conducted using SPSS Statistics 29 software (IBM SPSS, Inc., IBM, New York, NY, USA). Plots illustrating the concentrations of plasma cytokines and Hb were constructed using the R packages ggbreak v0.1.5 [38] and ggplot2 (R v.4.3.2).

## 3. Results

### 3.1. Patient Recruitment

The CONSORT flow diagram of the clinical trial is shown in Figure 1. One hundred sixty-two preterm infants were initially recruited, with fifty-nine excluded due to not meeting the inclusion criteria or declining participation. One hundred three preterm infants were ultimately included in the study, with fifty assigned to bLf and fifty-three to placebo. All participants received their allocated interventions. Follow-up data were collected for all participants; however, six from the bLf group and two from the placebo group discontinued the intervention due to various reasons, including NEC, digestive intolerance, or death. The ITT analysis included all randomized participants, with exclusions made only for specific secondary analyses due to processing issues.

### 3.2. Maternal and Newborn Characteristics

The gestational, perinatal, and neonatal characteristics are detailed in Table 2 and Table 3, with no significant differences between the study groups. No adverse events attributable to bLf or placebo administration were reported during the study period.

### 3.3. Incidence of LOS and Comorbidities Associated with Prematurity

As shown in Table 4, bLf supplementation reduced LOS by 46% vs. placebo (aRR, 0.54; 95% CI, 0.31–0.93; *p* = 0.028), corresponding to an NNT of 6. The incidence of BPD (any grade or moderate–severe), NEC, hsPDA, ROP, and brain injury at discharge was comparable between groups. Two deaths occurred in the bLf arm (one LOS-related); none in placebo (*p* > 0.05).

All organisms isolated from LOS episodes are listed in Appendix A. Distributions by individual taxa, probable source, and Gram category were comparable between groups (all *p* > 0.05).

### 3.4. Plasma Cytokines

Table 5 presents the baseline and final cytokine concentrations in both groups following the intervention. A time × intervention interaction emerged solely for MCP-1 (*p* = 0.022); it declined markedly in the placebo group (*p* < 0.001 ^b^) but remained unchanged under bLf (*p* > 0.05), resulting in higher MCP-1 levels in bLf-treated infants by the final intervention (*p* = 0.044 ^a^). After accounting for a higher baseline IL-8 level in the placebo arm (*p* = 0.043 ^a^), IL-8 decreased similarly in both groups (time *p* < 0.001), with no difference at the final intervention (*p* > 0.05). TNF-α levels were diminished only in the placebo recipients (*p* = 0.005 ^b^), whereas IL-6 and IL-10 levels decreased in both arms (time *p* < 0.001). IFN-γ levels increased over time in both groups (time *p* < 0.001), whereas IL-1β levels remained stable (time *p* > 0.05).

Despite the limited sample size and high variability, we aimed to determine whether the cytokine pattern was altered in the presence of LOS.

In LOS infants (Figure 2), a three-way interaction (time × intervention × LOS) was observed for IL-6 (*p* = 0.007, Figure 2A), and a borderline effect was noted for MCP-1 (*p* = 0.052, Figure 2B). Notably, in LOS patients receiving bLf, IL-6 levels remained stable, while MCP-1 levels increased, contrasting with the overall decline in these cytokine levels observed in the other groups. Further stratified LOS data obtained for IFN-γ, IL-10, IL-1β, IL-8, and TNF-α did not show any significant differences between groups; consequently, these are not presented here but are available upon request.

### 3.5. Antioxidant Enzyme Activities and TAC

Although the findings do not clearly demonstrate inter-group differences (time × intervention *p* > 0.05, Table 5), possibly limited by the sample size, variations in antioxidant enzyme activity were observed over time, particularly in CAT and TAC. CAT activity slightly decreased only in the bLf group (*p* = 0.019 ^b^), TAC remained stable after bLf administration but showed a reduction in the control group (*p* = 0.002 ^b^). No significant changes were observed for GR, GPx, or SOD. Due to the limited sample size for these outcomes, LOS stratification was not performed.

### 3.6. Hemoglobin Levels

Considering the challenges posed by frequent transfusions in this population, we assessed the hemoglobin levels to evaluate possible inter-group differences. Among infants who were non-transfused, post-intervention hemoglobin levels were significantly higher in the bLf group than in the placebo group (*p* = 0.028, Figure 3). Conversely, among transfused infants, no significant differences in hemoglobin levels were found between groups.

Nevertheless, no three-way time × intervention × RBCT interaction was detected.

## 4. Discussion

In this double-blind RCT, we assessed the effect of bLf on LOS, circulating cytokines, antioxidant status, and hemoglobin concentrations in very preterm and VLBW infants. After adjustment for GA, receipt of human milk (exclusive or partial), and prolonged intubation (≥48 h), bLf was associated with a 46% reduction in LOS, indicating that the protective effect remained after accounting for human milk exposure. This clinical effect paralleled a distinctive cytokine pattern, with persistently elevated MCP-1 levels in the bLf arm compared to the placebo group, suggesting a potential role of this chemokine in bLf’s immunomodulatory effects. Antioxidant data might indicate preservation of TAC with bLf, although the comparison with the control group based on the interaction analysis was not different. Furthermore, hemoglobin levels were higher in those infants receiving bLf and who had not undergone transfusions. These findings suggest that bLf may exert positive immunological and hematological effects in this population.

In the initial study protocol description, available on clinicaltrials.gov (NCT03472170), the primary outcome was to evaluate the efficacy of enteral administration of bLf in reducing probable and proven sepsis in preterm infants. However, in line with contemporary NICU, the primary endpoint is to maximize diagnostic specificity and cross-study comparability [39]. Our findings agree with several small trials and meta-analyses reporting lower LOS rates with bLf [14,15,16,18,19,20,40], likely due to its combined antimicrobial and immunomodulatory actions. However, these findings contrast with those of two large multicenter megatrials, ELFIN [12] and LIFT [13], that found no reduction in infection-related outcomes. Possible explanations include the higher incidence of LOS in our placebo arm (39.6% vs. 16.5% in ELFIN and 10.7% in LIFT), the lower proportion of infants receiving human milk in our unit (~72% vs. >94% in the megatrials), and the closer oversight inherent to a single-center design, which may have minimized variations in dosing and protocol adherence compared with multicenter trials. The pathogen spectrum, predominantly coagulase-negative staphylococci with fewer enteric organisms, indicated that infections primarily originated from catheter/skin sources rather than gut translocation, and it did not differ between study arms.

In our study, BPD was numerically less frequent in the bLf arm; however, consistent with previous investigations and meta-analyses [9,10,12,15,20,41,42,43], the difference compared to the reference group did not reach statistical significance. Any potential pulmonary benefit of bLf, therefore, remains speculative [44,45].

Both LOS and BPD are multifactorial conditions that overlap with other morbidities such as NEC, hsPDA, ROP, and brain injury. In line with other studies [9,12,13,20,41,43], we did not detect clear effects of bLf on these outcomes. The combination of multifaceted pathogenesis, their lower baseline incidence in contemporary NICUs, and our modest sample size, primarily powered to detect differences in LOS, probably limited our ability to detect smaller effects in these secondary outcomes.

The immunomodulatory capacity of Lf has been widely reported in vitro and in vivo, as it can bind and neutralize pathogen-associated molecular patterns, modulate cytokine production, and influence both innate and adaptive immune responses [23,46]. Our cytokine analyses revealed a pattern compatible with controlled immune maturation in the bLf group, providing a mechanistic context for the clinical findings.

Overall, and in accordance with our findings, cytokine profiles may indicate a maturing immune response in preterm infants [47,48]. Over time, levels of IL-6, a proinflammatory cytokine, and the chemokine IL-8 decreased, while IFN-γ increased, suggesting an enhanced T helper 1 response. Similarly, IL-10, which has anti-inflammatory and regulatory functions, tended to decline in both groups, potentially signaling a shift away from the early-life anti-inflammatory bias. In contrast, IL-1β levels, another proinflammatory cytokine, remained stable throughout the study period. These observations are consistent with findings by Lee and colleagues, who reported a comparable pattern of immune maturation [49].

The most distinctive finding was the differential modulation of MCP-1; its levels remained stable in the bLf-treated group while they declined in the placebo group. This pattern suggests that bLf exerts a regulatory influence, preserving the chemokine-driven recruitment of monocytes without triggering harmful hyperinflammation. Mechanistic studies support this view, indicating that bLf enhances monocyte survival and functional activation by inducing autocrine growth factors, such as GM-CSF and M-CSF, while also binding to microbial components like lipopolysaccharide. Consequently, bLf likely tempers excessive TLR-4-mediated signaling, preventing an inflammatory surge, yet maintains a sufficient MCP-1-mediated chemotactic response [50,51,52,53]. By contrast, the decline in MCP-1 in the placebo group may reflect premature downregulation of this chemokine axis, potentially impairing immune cell recruitment and increasing susceptibility to infection in these infants. Actually, subgroup analysis further highlighted the immunomodulatory effects of bLf in infants with LOS. While non-LOS infants exhibited the expected progressive declines in IL-6 and MCP-1 as part of normal immune maturation, those with LOS receiving bLf showed an increase in MCP-1 levels, with IL-6 remaining stable, unlike the placebo group, which exhibited significant declines in these cytokines without corresponding clinical benefit. This balanced cytokine profile may optimize pathogen clearance without provoking harmful hyperinflammation [51,54].

In addition to its immunomodulatory role, bLf may also influence antioxidant status. Preterm infants are particularly vulnerable to oxidative damage due to immature antioxidant defenses, infections, and prolonged oxygen therapy [55]. Given that Lf can bind iron, limiting the iron-driven formation of reactive oxygen species, it was plausible that bLf could preserve or enhance antioxidant capacity [23,56]. In our study, although sample attrition reduced statistical power for enzyme assays, antioxidant enzyme activity remained largely similar between groups, except for CAT, which slightly decreased in the bLf group. This possible decrease could align with transient iron sequestration, which may delay heme loading of new CAT, and with lower H_2_O_2_ levels potentially redirecting peroxide detoxification toward heme-independent systems, such as GPx and peroxiredoxins [57,58]. Regarding overall antioxidant status, TAC remained stable in the bLf group but declined in the placebo group. The preservation of TAC supports bLf’s iron-binding properties and suggests a net antioxidant advantage. These findings are consistent with bLf’s purported ability to limit oxidative damage, thereby mitigating tissue injury and improving outcomes [23,56]. Although evidence in preterm infants is still limited, these observations warrant further mechanistic and clinical investigation of bLf-mediated redox protection.

Beyond the infection-driven immune response, bLf supplementation was also linked to higher post-intervention hemoglobin levels in non-transfused infants, a finding that might suggest a possible role in erythropoiesis. Although the mechanisms remain unclear, bLf has been proposed to influence iron metabolism and hematopoiesis, potentially through its interaction with transferrin receptors and its ability to regulate inflammatory mediators involved in erythropoiesis. This aligns with findings from other studies, which indicate that bLf supplementation improves hemoglobin levels [22,59]. Nevertheless, our findings should be interpreted with caution, as they were based on a simple statistical test.

The present study has several limitations. First, the sample did not reach the expected size due to COVID-19-related recruitment interruptions and early termination of enrollment to minimize pandemic-related bias, and the trial was primarily powered to detect differences in LOS incidence. This may have reduced sensitivity for detecting differences in secondary endpoints, low-frequency events, and subgroup analyses. Second, our single-center design may restrict the broader applicability of our findings. Third, logistical constraints in blood collection and processing reduced the number of infants with analyzable antioxidant status data, potentially masking genuine differences. In the future, larger-scale studies should incorporate more robust antioxidant assessments to clarify these findings. Finally, the very low mortality rate prevented meaningful evaluation of any survival effect.

Despite its limitations, this study has several important strengths. It was conducted as a rigorous randomized trial with high protocol adherence and involved a carefully selected cohort of preterm infants born in a single hospital, ensuring consistency in clinical care. The study included comprehensive assessments of LOS, other neonatal morbidities, and key biological markers, despite the inherent challenges of working with a fragile population where sample volumes are limited and clinical conditions can change rapidly. Importantly, the integrative evaluation of immune, antioxidant, and hematological pathways provides valuable insights into the multifaceted actions of bLf. To our knowledge, this is a pioneering study suggesting that the protective effect of bLf against LOS may be mediated through immunomodulatory and antioxidant mechanisms in very preterm infants.

## 5. Conclusions

These findings strongly support the hypothesis that enteral supplementation with bLf may reduce the incidence of LOS in vulnerable preterm populations. The preserved antioxidant capacity and nuanced immunomodulation observed in the bLf group might further support a plausible multifactorial mechanism by which bLf helps protect against infection, inflammation, and oxidative injury. Continued research through larger, multicenter studies will be essential to confirm these findings, determine optimal dosing strategies, clarify long-term clinical outcomes, and establish bLf as a valuable adjunct in the care of preterm infants at high risk for serious neonatal complications.

## Figures and Tables

**Figure 1 nutrients-17-03154-f001:**
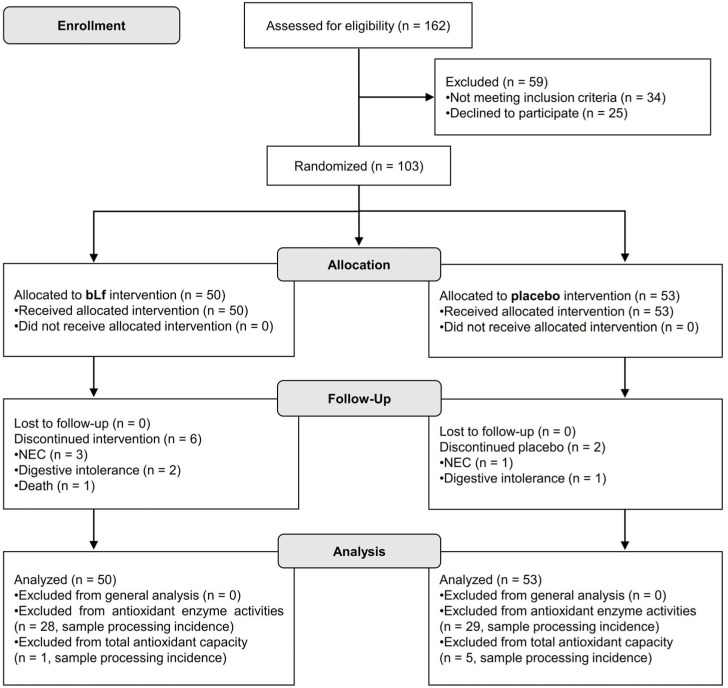
CONSORT flow diagram illustrating participant progression through each phase of the randomized clinical trial.

**Figure 2 nutrients-17-03154-f002:**
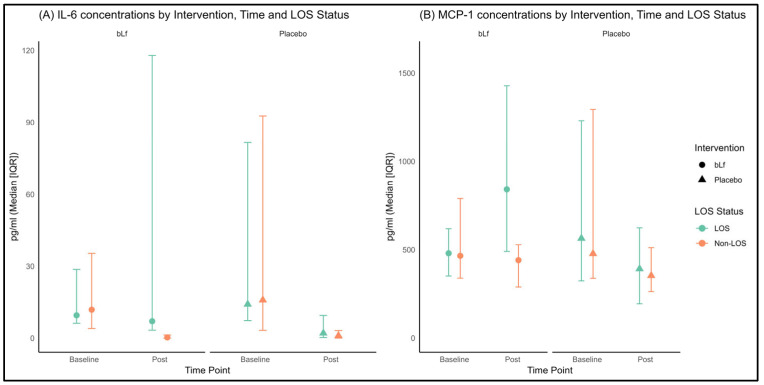
(**A**) Plasma concentrations of IL-6 at baseline and post-intervention, stratified by intervention group and LOS status; (**B**) Plasma concentrations of MCP-1 at baseline and post-intervention, stratified by intervention group and LOS status. The circles and triangles in the figure represent the intervention and control groups, respectively; the color is not used for group assignment. Color depends on LOS status, green indicates non-LOS and orange indicates LOS.

**Figure 3 nutrients-17-03154-f003:**
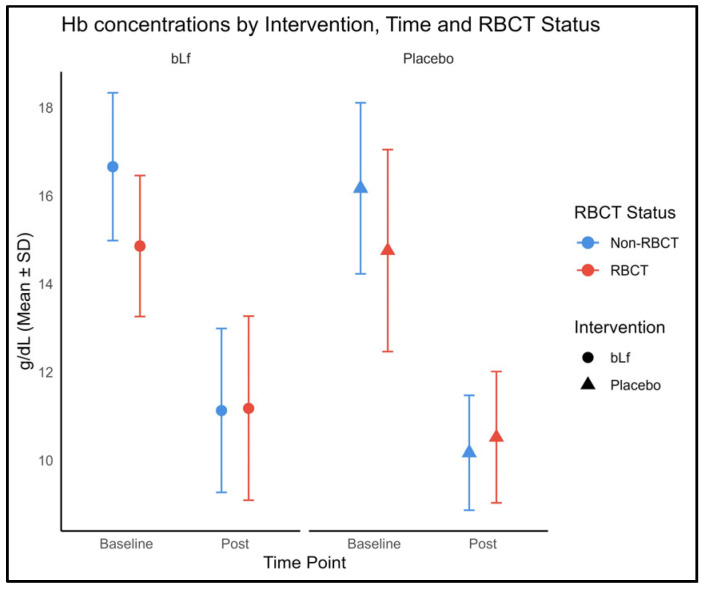
Hemoglobin concentrations at baseline and post-intervention, stratified by intervention group and RBCT status. The circles and triangles in the figure represent the intervention and control groups, respectively; the color is not used for group assignment. Color depends on RBCT status, blue indicates non-LOS and red indicates LOS.

**Table 1 nutrients-17-03154-t001:** Criteria for Selected Neonatal Morbidities.

Condition	Diagnostic Criteria	References *
NEC	Diagnosed according to the modified Bell staging criteria.	[32]
ROP	Classified with the International Classification of Retinopathy of Prematurity.	[33]
BPD	Dependency on supplemental oxygen for ≥28 days, with severity graded at 36 weeks post-menstrual age as mild (room air), moderate (<30% O_2_), or severe (≥30% O_2_ and/or positive-pressure support), per National Institute of Child Health and Human Development criteria.	[34]
hsPDA	Echocardiographic evidence of left-to-right shunting (ductal diameter > 1.5 mm, left-atrium–to-aortic-root > 1.5, and/or diastolic flow reversal in the descending aorta) plus clinical signs of pulmonary over-circulation and systemic hypoperfusion requiring medical or surgical intervention.	[35]
Brain injury	Pathologic findings on standardized cranial imaging (ultrasonography and/or magnetic resonance imaging) obtained at discharge.	-
Overall mortality	All causes of mortality during hospitalization.	-
Mortality due to LOS	Any death occurring after a LOS episode in the absence of other evident causes.	-

BPD, bronchopulmonary dysplasia; hsPDA, hemodynamically significant patent ductus arteriosus; LOS, late onset neonatal sepsis; NEC, necrotizing enterocolitis; ROP, retinopathy of prematurity. * Diagnostic criteria compiled from references [32,33,34,35].

**Table 2 nutrients-17-03154-t002:** Gestational-perinatal data by intervention group.

	bLf (*n* = 50)	Placebo (*n* = 53)	*p* Value
Maternal age, years	33.82 ± 5.14	33.81 ± 4.01	0.992
Preeclampsia	11 (22)	10 (18.9)	0.693
Maternal diabetes	2 (4)	4 (7.5)	0.679
Maternal chorioamnionitis	5 (10)	9 (17)	0.301
Complete course of antenatal corticosteroid	36 (72)	40 (75.5)	0.689
Maternal antibiotic administration	25 (50)	24 (45.3)	0.632
Vertical sepsis risk factors	21 (42)	23 (43.4)	0.886

Data are presented as *n* (%) for categorical variables or mean ± SD for continuous variables. Categorical variables were analyzed using the chi-square test, and continuous variables were compared using Student’s *t*-test. bLf, bovine lactoferrin.

**Table 3 nutrients-17-03154-t003:** Neonatal data by intervention group.

	bLf (*n* = 50)	Placebo (*n* = 53)	*p* Value
General characteristics			
GA, weeks	30.04 ± 2.12	30.15 ± 2.23	0.797
GA ≤ 29 weeks	17 (34)	18 (34)	0.997
Sex, male	29 (58)	25 (47.2)	0.271
Birth weight, g	1357.02 ± 312.53	1298.75 ± 289.51	0.328
Birth weight, z-score	−0.28 ± 0.90	−0.52 ± 0.95	0.178
Birth length, cm	39.23± 3.19	39.08 ± 3.46	0.820
Birth length, z-score	−0.10 ± 1.00	−0.22 ± 1.15	0.574
Head circumference at birth, cm	27.61 ± 2.47	27.47 ± 2.00	0.754
Head circumference at birth, z-score	−0.05 ± 1.12	−0.17 ± 1.15	0.598
Therapeutic interventions			
Surfactant	30 (60)	30 (56.6)	0.842
Intubation	26 (52)	21 (39.60)	0.208
Intubation ≥ 48 h	14 (28)	12 (22.6)	0.532
Duration of respiratory support, days	18.94 ± 25.83	19.17 ± 22.39	0.961
Blood product transfusion	22 (44)	19 (35.8)	0.398
Postnatal antibiotic use	31 (62)	33 (62.3)	0.978
PPIs use	22 (44)	23 (43.4)	0.951
Nutritional characteristics			
Duration of parenteral nutrition, days	15.14 ± 13.76	15.04 ± 14.90	0.971
Time to first feed, days	1.84 ± 1.14	1.57 ± 0.89	0.180
Time to achieve full enteral feeding, days	16.75 ± 12.32	16.68 ± 11.27	0.976
Complete enteral feeding with exclusively own mother’s milk or mixed feeding *	37 (77.1)	38 (71.7)	0.503
Characteristics at discharge ^†^			
Total hospital stay, days	49.40 ± 22.86	50.43 ± 20.91	0.811
Weight at discharge, g	2612.65 ± 400.96	2581.28 ± 320.93	0.664
Weight at discharge, z-score	−1.00 ± 0.86	−1.01 ± 1.06	0.970
Length at discharge, cm	47.10 ± 2.30	47.24 ± 2.31	0.762
Length at discharge, z-score	−0.68 ± 0.93	−0.58 ± 1.29	0.667
Head circumference at discharge, cm	33.13 ± 1.02	33.32 ± 1.12	0.371
Head circumference at discharge, z-score	−0.33 ± 0.75	−0.17 ± 0.90	0.362

Data expressed as *n* (%) for categorical variables or mean ± SD for continuous variables. The χ^2^ test was used to compare categorical variables, and Student’s *t*-test was used to compare continuous quantitative variables. * Percentages for the bLf group are based on *n* = 48; two infants died before full enteral feeding. ^†^ Values for the bLf group are calculated on the survivors at discharge (*n* = 48). bLf, bovine lactoferrin; PPIs, Proton pump inhibitor.

**Table 4 nutrients-17-03154-t004:** Comparison of incidence between the primary and secondary variables under study by intervention group.

	bLf	Placebo	Crude RR with bLf (95% CI)	aRR (95% CI) ^a^	*p* Value ^b^
LOS	11/50 (22)	21/53 (39.6)	0.55 (0.299–1.031)	0.536 (0.308–0.934)	**0.028**
BPD	9/48 (18.8)	17/53 (32.1)	0.585 (0.288–1.186)	0.695 (0.398–1.214)	0.201
BPD moderate/severe	4/48 (8.3)	4/53 (7.5)	1.104 (0.292–4.174)	-	1 ^†^
NEC	3/50 (6)	2/53 (3.8)	1.590 (0.277–9.122)	-	0.672 ^†^
hsPDA	8/50 (16)	9/53 (17)	0.942 (0.395–2.250)	-	0.893
ROP	1/48 (2.1)	2/53 (3.8)	0.552 (0.052–5.897)	-	1 ^†^
Brain injury at discharge	2/50 (4)	4/53 (7.5)	0.530 (0.102–2.767)	-	0.679 ^†^
Mortality	2/50 (4)	0/53 (0)	-	-	0.233 ^†^
Mortality due to LOS	1/50 (2)	0/53 (0)	-	-	0.485 ^†^

Data expressed as frequency/total number (%) for each arm. Crude RR (95% CI) and aRR (95% CI) are shown. Bold values are significant (*p* < 0.05). ^a^ aRR estimated with a modified Poisson regression (log link, robust variance), adjusting for GA at birth (weeks), feeding type (exclusively human milk or mixed feeding vs. formula only), and ventilatory support (LOS model: endotracheal intubation > 48 h; BPD model: total days of invasive respiratory support until BPD diagnosis criteria were met). Adjustments were performed only for outcomes with ≥25 total events to ensure model stability. ^b^ When an adjusted model could not be fitted, the χ^2^ test was used; ^†^ indicates Fisher’s exact test (when expected cell counts were <5). RR could not be calculated for mortality outcomes due to the absence of events in the placebo group. aRR, adjusted relative risk; bLf, bovine lactoferrin; BPD, bronchopulmonary dysplasia; CI, confidence interval; GA, gestational age; hsPDA, hemodynamically significant patent ductus arteriosus; LOS, late-onset neonatal sepsis; NEC, necrotizing enterocolitis; ROP, retinopathy of prematurity; RR, relative risk.

**Table 5 nutrients-17-03154-t005:** Effect of Time and Intervention on Cytokine levels, Antioxidant Enzyme Activities and TAC.

		bLf			Placebo		*p* Value	
	N	Baseline	Final Intervention	N	Baseline	Final Intervention	Time	T × I
**Cytokines**								
IFNγ, pg/mL	50	3.58 [0.61–10.87]	16.03 [8.21–30.04] ^b^	53	0.61 [0.61–12.51]	13.74 [5.63–35.5] ^b^	**<0.001**	0.502
IL10, pg/mL	50	34.71 [18.61–122.62]	15.04 [9.29–26.26] ^b^	53	40.27 [21.06–132.54]	14.61 [9.65–23.24] ^b^	**<0.001**	0.497
IL1β, pg/mL	50	0.58 [0.01–1.60]	1.59 [0.29–3.01]	53	0.68 [0.01–3.59]	1.55 [0.67–2.64]	0.137	0.564
IL6, pg/mL	50	11.48 [4.81–29.57]	0.66 [0.03–3.19] ^b^	53	14.14 [4.51–81.65]	1.34 [0.03–5.07] ^b^	**<0.001**	0.601
IL8, pg/mL	50	59.38 [29.11–106.80]	26.28 [13.79–76.79] ^b^	53	101.4 [45.73–291.81] ^a^	22.81 [12.36–67.46] ^b^	**<0.001**	0.073
TNFα, pg/mL	50	31.52 [22.77- 48.42]	29.13 [20.64–42.63]	53	36.82 [24.88–44.44]	26.33 [18.81–41.83] ^b^	**0.004**	0.276
MCP-1, pg/mL	50	466.90 [348.31–758.67]	457.95 [304.43–581.16]	53	544.91 [333.16–1251.92]	362.5 [257.73–524.18] ^a,b^	**<0.001**	**0.022**
**Antioxidant Enzyme Activities and TAC**								
GR, U enzyme/g Hb	22	1.27 [1.01–1.67]	1.28 [1.07–1.58]	24	1.11 [0.77–1.64]	1.31 [0.96–1.65]	0.290	0.715
GPx, U enzyme/g Hb	23	27.38 [19.35–35.57]	29.76 [18.90–35.27]	24	25.89 [19.98–28.31]	24.18 [17.26–30.36]	0.871	0.131
SOD, U enzyme/mg Hb	23	0.11 [0.07–0.17]	0.12 [0.09–0.14]	24	0.10 [0.08–0.13]	0.11 [0.08–0.14]	0.400	0.213
CAT, U enzyme/mg Hb	23	1.05 [0.98–1.08]	1.01 [0.94–1.06] ^b^	24	1.06 [1.01–1.10]	1.05 [0.98–1.09]	**0.007**	0.528
TAC, mmol of Trolox/L of cell lysate	49	0.88 [0.57–1.00]	0.85 [0.69–0.96]	48	0.91 [0.78–1.05]	0.78 [0.62–0.96] ^b^	**0.006**	0.108

Data were transformed using a base-10 log scale to reduce dispersion; however, results are presented using untransformed values expressed as medians and interquartile ranges for clarity and ease of interpretation. Bold values are significant (*p* < 0.05). ^a^ *p* < 0.05 indicates statistical significance between intervention groups (bLf vs. placebo) at the same point (baseline or final intervention). For IL-8, ANCOVA was conducted to adjust for baseline differences between groups. The resulting *p* = 0.572 shows no significant difference between the placebo and bLf at the final intervention point. ^b^ *p* < 0.05 denotes statistical significance between time points (baseline vs. final intervention) within the same intervention group (bLf or placebo). ANCOVA, analysis of covariance; bLf, bovine lactoferrin; CAT, catalase; GPx, Glutathione Peroxidase; GR, Glutathione Reductase; IFNγ, interferon gamma; IL, interleukin; MCP-1, monocyte chemotactic protein; SOD, Superoxide Dismutase; TAC, Total Antioxidant Capacity; T × I, time × intervention; TNFα, tumor necrosis factor alpha.

## Data Availability

The datasets generated and analyzed during the current study are not publicly available due to ethical and regulatory constraints. Specifically, the families who participated provided informed consent allowing data use exclusively by the original research team. Sharing these data more broadly could compromise participant confidentiality and exceed the scope of consent. However, researchers with a reasonable request may contact the corresponding author to explore the possibility of providing encoded data under appropriate ethical oversight.

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
