# Peer review of "Preventing Sepsis in Preterm Infants with Bovine Lactoferrin: A Randomized Trial Exploring Immune and Antioxidant Effects"

_nutrients, 2025, doi:10.3390/nu17193154_

Round 1
Reviewer 1 Report
Comments and Suggestions for Authors
Major comments: The authors presented an extremely well conducted an excellent double blind, randomized controlled trial of bovine lactoferrin in prevention of late onset neonatal sepsis.
The authors could further improve the introduction by providing detailed information of the effects of bovine versus human lactoferrin on bacterial infection.
In the result section the authors should present for each group perhaps in a separate table the list of all organisms isolated in intervention and placebo groups and analyse the difference (s). The authors could then discuss in the discussion section the likely origin of the organisms found: for example gut versus plastic catheters in each group.
Minor comments:
Abstract: The sentence "post-hoc, the placebo group declined (p = 0.002), while bLf remained stable (p = 0.400)" is unclear: Does it refer to TAC?
Author Response
Response: Thank you very much for your effort in reviewing our manuscript and for the suggestions to be improved. We have been working on our manuscript in accordance with your suggestions. We believe that this has helped increase the quality of the paper and hope it meets your expectations. All the specific information required has been included throughout the manuscript using “track changes”. Please find below our point-by-point response to the questions raised. The number of lines specified indicates the location of the modified text in the revised version after using “track changes”.
Question 1: The authors could further improve the introduction by providing detailed information of the effects of bovine versus human lactoferrin on bacterial infections.
Response 1: Thank you for your observation. As recommended, we have included additional information about this in the Introduction section (lines 66-79): “Both human and bovine lactoferrin (hLf/bLf) inhibit bacteria via iron sequestration and direct membrane/lipopolysaccharide (LPS) interaction, sharing the same N-terminal LPS-binding region. Upon pepsin digestion, N-terminal fragments, such as lactoferricin, are released. In vitro, bovine-derived peptides exhibit minimum inhibitory concentrations four times lower than their human analogue, with activity being highest in the apo form and modulated by divalent cations, although effects are strain- and context-dependent (Gruden, 2021; Vorland 1998). Beyond its similarity to hLf, bLf is manufactured on an industrial scale under Current Good Manufacturing Practice (cGMP), and its use in foods is supported by regulatory assessments from the U.S. Food and Drug Administration (FDA; GRAS ‘no-questions’) and the European Food Safety Authority (EFSA; novel-food opinions). Moreover, bLf is more readily available commercially and less costly than recombinant hLf, making it the primary option for investigation and nutritional supplementation, especially in preterm infants (Wakabayashi, 2018)”.
Question 2: In the result section the authors should present for each group perhaps in a separate table the list of all organisms isolated in intervention and placebo groups and analyse the difference (s). The authors could then discuss in the discussion section the likely origin of the organisms found: for example gut versus plastic catheters in each group.
Response 2: Thank you for this helpful suggestion. We created a supplementary table (Supplementary Material; Table S2) listing, by randomized group, all organisms isolated in LOS episodes, including Gram classification and presumed origin. We compared distributions using Fisher’s exact tests for 2×2 contrasts (individual taxa) and the Fisher-Freeman–Halton exact test for 2×3 tables (probable source and Gram category). In the Results section, we have included a summary of this table (lines 330-332) as follows: “All organisms isolated from LOS episodes are listed in Supplementary Table S2. Distributions by individual taxa, probable source, and Gram category were comparable between groups (all p > 0.05)”.
In the Discussion section, we have commented on the likely origin of the isolates (catheter/skin flora vs enteric organisms), noting that the spectrum did not differ between the study arms (lines 403-405): “The pathogen spectrum, predominantly coagulase-negative staphylococci with fewer enteric organisms, indicated that infections primarily originated from catheter/skin sources rather than gut translocation, and it did not differ between study arms”.
Question 3: Minor comments: Abstract: The sentence "post-hoc, the placebo group declined (p = 0.002), while bLf remained stable (p = 0.400)" is unclear: Does it refer to TAC?
Response 3: We appreciate this helpful comment and apologize for the lack of clarity. The sentence refers to TAC. We have revised the abstract to make this more explicit and avoid ambiguity.
Changes in Abstract (lines 43-44): “Although TAC showed a non-significant interaction, the placebo group declined (p = 0.002), while bLf remained stable (p = 0.400) in the post hoc analysis”.
References mentioned above
- Gruden Š, Poklar Ulrih N. Diverse mechanisms of antimicrobial activities of lactoferrins, lactoferricins, and other lactoferrin-derived peptides. Int J Mol Sci. 2021;22(20):11264. doi: 10.3390/ijms222011264.
- Vorland LH, Ulvatne H, Andersen J, Haukland HH, Rekdal Ø, Svendsen JS, et al. Lactoferricin of Bovine Origin is More Active than Lactoferricins of Human, Murine and Caprine Origin. Scand J Infect Dis. 1998;30(5):513–7. doi: 10.1080/00365549850161557.
- Wakabayashi H, Yamauchi K, Abe F. Quality control of commercial bovine lactoferrin. Biometals. 2018 Jun;31(3):313–319. doi:10.1007/s10534-018-0098-2.

Reviewer 2 Report
Comments and Suggestions for Authors
I have read this paper with a background on perinatal clinical research, including strategies to improve LOS outcome or events. The current paper further builds on the hypothesis of the protective effect of lactoferrin.
When compared to the initial protocol description as provided at clinicaltrials.gov, it seems that the primary outcome has been adapted from probable and culture proven to culture proven infection, unless I have misunderstood the statement. Since the primary outcome has been used to power the study, it is important to reconsider this, and to provide the data based on the initial primary outcome. It is still likely useful to explore additional secondary outcomes, like culture proven LOS, but based on the protocol, the current reporting has shortages. Related to this, the current 2.6.1. seems not to be in line with what is described in the clinicaltrials.gov website. This is my main concern on the current version of the paper.
I might have missed this, but when was lactoferrin initiated (? Volume dependent, you suggest at 72 h), and when was it stopped ? Related to this, what this discharge home, or transfer ? and how was the enteral nutritional strategy of the units involved.
Related to this, have you explored the potential impact of inflammatory markers with gestational age, or postnatal age ?
Author Response
REVIEWER REPORT
Response: We sincerely appreciate your thorough review and valued insights. It is an honor to receive feedback from a reviewer with extensive experience in perinatal clinical research. All the specific information required has been included throughout the manuscript using “track changes”. Please find below our point-by-point response to the questions raised. The number of lines specified indicates the location of the modified text in the revised version using “track changes”.
Question 1: When compared to the initial protocol description as provided at clinicaltrials.gov, it seems that the primary outcome has been adapted from probable and culture proven to culture proven infection, unless I have misunderstood the statement. Since the primary outcome has been used to power the study, it is important to reconsider this, and to provide the data based on the initial primary outcome. It is still likely useful to explore additional secondary outcomes, like culture proven LOS, but based on the protocol, the current reporting has shortages. Related to this, the current 2.6.1. seems not to be in line with what is described in the clinicaltrials.gov website. This is my main concern on the current version of the paper.
Response 1: We appreciate the opportunity to clarify this issue. Although the 2016 protocol listed “probable and proven” LOS, we report culture-confirmed LOS as the primary endpoint to maximize diagnostic specificity and cross-study comparability, consistent with contemporary Spanish NICU surveillance practice (e.g., Grupo Castrillo/SEN1500). Methodological reviews emphasize substantial heterogeneity in neonatal sepsis definitions and recommend standardized, objective endpoints. Culture-proven infection is the most reproducible for trials (McGovern, 2020; Hayes, 2023). In line with current surveillance practice, large networks likewise operationalize LOS as culture-confirmed (Flannery, 2022). This approach is consistent with the field’s precedent (Manzoni, 2009; Akin, 2014; Dai, 2015; Barrington, 2016; Liu, 2016; Sherman, 2016; LIFT, 2020) defined LOS as culture-proven. Recent syntheses (e.g., Wang et al.; JAMA Pediatrics, 2023) similarly prioritize culture-proven LOS as the primary and comparable endpoint.
Only three additional episodes met criteria for probable (culture-negative) LOS (two in bLf; one in placebo); the inclusion of them does not change the direction or significance of the effect (aRR: 0.57, IC 95% 0.33–0.98; p = 0.042). Our conclusions, therefore, rest on the endpoint most consistent with national surveillance standards and our unit’s baseline data. Should the Editor deem it necessary, we can provide these data in supplementary material.
It is also worth noting that COVID-19 led to unavoidable recruitment interruptions and altered care pathways. To preserve safety and protocol fidelity under changed sociosanitary conditions, we did not reach the planned sample. We explicitly acknowledge the resulting loss of power in the manuscript and report an approximate post-hoc power of ~70% for the observed effect size.
Manuscript changes: We have added a clarifying paragraph in the Discussion section (lines 390-394), explaining the rationale for a culture-confirmed primary outcome: “In the initial study protocol description, available on clinicaltrials.gov (NCT03472170), the primary outcome was to evaluate the efficacy of enteral administration of bLf in the reduction of probable and proven sepsis in preterm infants. However, in line with contemporary NICU surveillance and methodological guidance, we report culture-confirmed LOS as the primary endpoint to maximize diagnostic specificity and cross-study comparability”. In the Methods section (Sample size, subsection 2.7, lines 258–261), we now state the reason for not reaching the target: “However, COVID-19 caused unavoidable recruitment interruptions and changes in care pathways, limiting enrollment to 103 preterm infants (50 in the bLf arm and 53 in the placebo arm). A post hoc power analysis indicated approximately 70% power to detect the estimated effect size”. Additionally, in the Discussion section (Limitations, lines 475-476), we have added: “First, the sample did not reach the expected size due to COVID-19-related recruitment interruptions and early termination of enrollment to minimize pandemic-related bias”.
Response 2: We appreciate this helpful comment and apologize that the original version was not sufficiently clear. As detailed in the revised Methods section (Intervention, subsection 2.4):
- Initiation: bLf supplementation began within the first 72 hours of life (EXISTING; lines 166-167).
- Scope of initiation: regardless of the volume of enteral feeds achieved, and was also given during periods of nil per os (NPO) [NEW; lines 167-168].
- Continuation/stop rule: it was continued once daily for up to four weeks or until hospital discharge to home, whichever occurred first (clarified wording; lines 168-169).
- Nutritional strategy: “Our NICU followed a standardized feeding protocol based on the current recommendations of the European Society for Paediatric Gastroenterology Hepatology and Nutrition and the Spanish Society of Neonatology on nutrient intakes and nutritional management for preterm infants. The nutritional protocol included: 1) Administration of standardized parenteral nutrition with all macronutrients within the first hours of life; 2) Early trophic enteral feedings; 3) Gradual advancement of enteral volumes; 4) Preference for mother’s own milk; donor milk was not available during the study period, and formula was used when necessary; 5) Fortification of breastfeeding starting at an approximate volume of 80 ml/kg/d; 6) Discontinuation of parenteral nutrition when enteral nutrition reached a volume of 100-120 ml/kg/d for two days. Probiotics were not administered”; [NEW; lines 170-180].
For completeness in addressing the reviewer’s comment (not included in the manuscript text): no infants were transferred to other hospitals or units during the study; one infant was discharged home before day twenty-eight, and supplementation ended at discharge in that case.
Question 3: Related to this, have you explored the potential impact of inflammatory markers with gestational age, or postnatal age?
Response 3: We appreciate this thoughtful question. We have examined whether cytokine concentrations were associated with gestational age. At baseline (all samples collected within the first seventy-two hours of life), Spearman correlations revealed no statistically significant relationships between gestational age and cytokine levels (Response Table question 3). Given the absence of correlation and the comparable gestational ages and perinatal characteristics between randomized groups, we did not pursue additional age-focused analyses.
|
Response Table question 3. Spearman correlations between GA and baseline cytokine concentrations. |
||
|
Cytokines at baseline (n=103) |
ρ (Spearman) with GA |
p-value |
|
IFNγ, pg/ml |
-0.191 |
0.053 |
|
IL10, pg/ml |
-0.116 |
0.242 |
|
IL1β, pg/ml |
-0.041 |
0.684 |
|
IL6, pg/ml |
-0.068 |
0.497 |
|
IL8, pg/ml |
-0.169 |
0.089 |
|
TNFα, pg/ml |
-0.103 |
0.301 |
|
MCP-1, pg/ml |
-0.037 |
0.707 |
|
ρ = Spearman’s rank correlation coefficient; GA = gestational age. Baseline samples collected within the first seventy-two hours of life. All tests are two-tailed. |
||
References mentioned above
- Akin IM, Atasay B, Dogu F, Okulu E, Arsan S, Karatas HD, et al. Oral lactoferrin to prevent nosocomial sepsis and necrotizing enterocolitis of premature neonates and effect on T-regulatory cells. Am J Perinatol. 2014;31(12):1111–20. doi: 10.1055/s-0034-1371704.
- Dai JZ, Xie C. The effect of lactoferrin supplementation combining Lactobacillus rhamnosus for prevention of late-onset sepsis in premature neonates. China Practical Medicine. 2015;10:98-100.
- Barrington KJ, Assaad MA, Janvier A. The Lacuna Trial: a double-blind randomized controlled pilot trial of lactoferrin supplementation in the very preterm infant. J Perinatol. 2016;36(8):666-9. doi:10.1038/jp.2016.24.
- Fernandez Colomer B, Cernada Badia M, Coto Cotallo D, Lopez Sastre J. The Spanish National Network Grupo Castrillo: 22 Years of Nationwide Neonatal Infection Surveillance. Am J Perinatol. 2020;37(Suppl 02):S71–5. doi: 10.1055/s-0040-1714256.
- Flannery DD, Edwards EM, Coggins SA, Horbar JD, Puopolo KM. Late-onset sepsis among very preterm infants. Pediatrics. 2022;150(6):e2022058813. doi:10.1542/peds.2022-058813.
- Hayes R, Hartnett J, Semova G, Murray C, Murphy K, Carroll L, et al. Neonatal sepsis definitions from randomised clinical trials. Pediatr Res. 2023;93(5):1141–1148. doi:10.1038/s41390-021-01749-3.
- Manzoni P, Rinaldi M, Cattani S, Pugni L, Romeo MG, Messner H, et al. Bovine Lactoferrin Supplementation for Prevention of Late-Onset Sepsis in Very Low-Birth-Weight Neonates: A Randomized Trial. JAMA. 2009;302(13):1421–8. doi: 10.1001/jama.2009.1403.
- McGovern M, Giannoni E, Kuester H, Turner MA, van den Hoogen A, Bliss JM, et al. Challenges in developing a consensus definition of neonatal sepsis. Pediatr Res. 2020;88(1):14–26. doi:10.1038/s41390-020-0785-x.
- Liu YH, Guan HS, Liang GJ, Li YZ, Zuo A. The effect of lactoferrin on low birth weight neonates during hospitalization. Maternal and Child Health Care of China. 2016;31(21):4464–5.
- Sherman MP, Adamkin DH, Niklas V, Radmacher P, Sherman J, Wertheimer F, et al. Randomized controlled trial of talactoferrin oral solution in preterm infants. J Pediatr. 2016;175:68–73. doi:10.1016/j.jpeds.2016.04.084.
- Tarnow-Mordi WO, Abdel-Latif ME, Martin A, Pammi M, Robledo K, Manzoni P, et al. The effect of lactoferrin supplementation on death or major morbidity in very low birthweight infants (LIFT): a multicentre, double-blind, randomised controlled trial. Lancet Child Adolesc Health. 2020;4(6):444–54. doi: 10.1016/S2352-4642(20)30069-7.
- Wang Y, Florez ID, Morgan RL, Foroutan F, Chang Y, Crandon HN, et al. Probiotics, Prebiotics, Lactoferrin, and Combination Products for Prevention of Mortality and Morbidity in Preterm Infants: A Systematic Review and Network Meta-Analysis. JAMA Pediatr. 2023;177(11):1158–67. doi:10.1001/jamapediatrics.2023.3849.

Round 2
Reviewer 2 Report
Comments and Suggestions for Authors
that the revisions are accurate, and suggest to accept